# Audio Deepfake Detection via a Fuzzy Dual-Path Time-Frequency Attention Network

**DOI:** 10.3390/s25247608

**Published:** 2025-12-15

**Authors:** Jinzi Li, Hexu Wang, Fei Xie, Xiaozhou Feng, Jiayao Chen, Jindong Liu, Juan Wang

**Affiliations:** 1Xi’an Key Laboratory of Human–Machine Integration and Control Technology for Intelligent Rehabilitation, Xijing University, Xi’an 710123, China; 2308540606022@stu.xijing.edu.cn (J.L.); wanghexu@xijing.edu.cn (H.W.); 2School of Information Science and Technology, Northwest University, Xi’an 710100, China; 202310355@stumail.nwu.edu.cn; 3Academy of Advanced Interdisciplinary Research, Xidian University, Xi’an 710071, China; fxie@xidian.edu.cn; 4School of Basic Sciences, Xi’an Technological University, Xi’an 710021, China; flxzfxz8@163.com (X.F.); chenjiayao202009@163.com (J.C.)

**Keywords:** audio deepfake detection, Pythagorean hesitant fuzzy sets, attention mechanism, time-frequency path

## Abstract

With the rapid advancement of speech synthesis and voice conversion technologies, audio deepfake techniques have posed serious threats to information security. Existing detection methods often lack robustness when confronted with environmental noise, signal compression, and ambiguous fake features, making it difficult to effectively identify highly concealed fake audio. To address this issue, this paper proposes a Dual-Path Time-Frequency Attention Network (DPTFAN) based on Pythagorean Hesitant Fuzzy Sets (PHFS), which dynamically characterizes the reliability and ambiguity of fake features through uncertainty modeling. It introduces a dual-path attention mechanism in both time and frequency domains to enhance feature representation and discriminative capability. Additionally, a Lightweight Fuzzy Branch Network (LFBN) is designed to achieve explicit enhancement of ambiguous features, improving performance while maintaining computational efficiency. On the ASVspoof 2019 LA dataset, the proposed method achieves an accuracy of 98.94%, and on the FoR (Fake or Real) dataset, it reaches an accuracy of 99.40%, significantly outperforming existing mainstream methods and demonstrating excellent detection performance and robustness.

## 1. Introduction

With the rapid advancement of speech synthesis (TTS), voice conversion (VC), and voice cloning technologies, audio deepfake techniques have become increasingly sophisticated. Forged audio has raised serious security concerns in fields such as entertainment, finance, and judiciary [1]. The goal of audio deepfake detection is to accurately distinguish forged audio from genuine audio, thereby safeguarding information security and social trust. However, as forgery techniques continue to improve, forged traces have become more concealed. Coupled with factors such as environmental noise and signal compression, the detection task faces significant challenges.

In recent years, both academia and industry have conducted extensive research on audio deepfake detection, leading to the development of various effective methods. Competitions such as the audio security challenges in ICASSP and Interspeech have facilitated the establishment of standardized datasets and evaluation metrics, promoting technological iteration. Representative studies include Dinkel et al. [2], who proposed an end-to-end anti-spoofing model based on raw waveforms, significantly improving detection accuracy; Ling et al. [3], who enhanced the model’s sensitivity to complex forged features by introducing multi-scale time-frequency representations; and Wang et al. [4], who improved the model’s cross-environment robustness by incorporating Multi-scale Permutation Entropy (MPE) to measure the complexity and dynamic characteristics of audio signals at different scales. Additionally, the Transformer architecture and its attention mechanism have been widely applied in audio forgery detection. For instance, Zaman et al. [5] further improved detection performance by introducing a Hybrid Transformer, achieving 99.55% training accuracy and 94.04% testing accuracy. Nevertheless, most existing methods overlook the feature uncertainty introduced by noise and compression, making it difficult to effectively capture the ambiguity and dynamic evolution of forged traces.

Although deep neural networks possess a certain degree of generalization ability against noise and disturbances, existing audio forgery detection methods usually treat Mel spectrograms and other time-frequency features as completely deterministic inputs. They do not explicitly model the degradation of feature reliability and the increase in uncertainty that are prevalent in real-world scenarios, such as compression, coding distortion, re-recording, and background noise. Under these circumstances, forgery traces are often significantly weakened or partially concealed, resulting in input features that simultaneously contain both “reliable regions” and “ambiguous or damaged regions”. Due to the lack of an uncertainty-aware mechanism, the model often fails to distinguish between the two and may mistakenly learn unreliable regions as normal features, thereby leading to a noticeable decline in generalization ability under unknown forgery methods or degraded conditions.

To address the aforementioned issues, this paper proposes a Dual-Path Time-Frequency Attention Network (DPTFAN) based on Pythagorean Hesitant Fuzzy Sets (PHFS). This model utilizes fuzzy set theory to dynamically characterize uncertainty in audio features and combines attention mechanisms along time and frequency paths to achieve deep fusion of time–frequency information, effectively enhancing the performance and robustness of audio deepfake detection. The main contributions of this paper are as follows:Uncertainty Modeling of Features Based on Pythagorean Hesitant Fuzzy Sets (PHFS): The Pythagorean Hesitant Fuzzy Set theory is introduced, utilizing ternary features of membership, non-membership, and hesitancy degrees to dynamically characterize the reliability and uncertainty in audio forgery features. This method not only provides a more detailed representation of the ambiguity in sample features but also effectively adapts to complex and variable environmental noise and forgery traces, thereby significantly enhancing the model’s robustness and generalization capability in practical applications.Design of a Lightweight Fuzzy Branch Network (LFBN): A lightweight fuzzy branch network is constructed, combined with high-level semantic features extracted from a pre-trained ResNet, to achieve fuzzy enhancement in the feature space. This module explicitly encodes the ambiguity and uncertainty of features, providing richer and more discriminative semantic information for subsequent networks. It improves the feature representation of forged audio while maintaining computational efficiency.Dual-Path Time-Frequency Attention Fusion Mechanism (DPTFAN): A dual-path attention network is designed, with parallel time and frequency paths focusing on the temporal dynamics and spectral structure of audio, respectively. By introducing dynamic adjustment based on fuzzy hesitancy degree weights, deep fusion of multi-dimensional information is achieved, effectively capturing subtle differences and implicit features in forged audio. This mechanism significantly enhances the model’s discriminative capability and detection stability, advancing the performance of audio deepfake detection.

## 2. Related Work

### 2.1. Audio Deepfake Detection Task

In recent years, with continuous breakthroughs in generative technologies such as Voice Conversion (VC), Text-to-Speech (TTS), and Voice Cloning, audio deepfake has become a critical security challenge urgently requiring solutions in the field of Artificial Intelligence Generated Content (AIGC). Forged audio not only exhibits high auditory deception but can also be used in attack scenarios such as identity impersonation and forged voice commands, posing serious security threats. Consequently, building robust and generalizable audio deepfake detection models has become a key research focus.

Early research primarily concentrated on traditional feature engineering methods, relying on handcrafted audio descriptors and statistical modeling. For example, Kinnunen et al. [6] introduced Constant Q Cepstral Coefficients (CQCC) and Gaussian Mixture Models (GMM) for spoofed speech detection in the ASVspoof 2015 challenge. While such methods exhibit low computational complexity, their detection accuracy significantly declines when faced with deep learning-generated speech signals.

With the rapid development of deep learning techniques, researchers have begun exploring end-to-end modeling approaches. Alzantot et al. [7] constructed a spectrogram-based classification network using CNNs, introducing the first end-to-end deepfake detection framework. Lavrentyeva et al. [8] systematically compared the combined performance of multi-channel acoustic features (such as LFCC, CQCC, and MFCC) with deep neural network architectures in ASVspoof 2019 and proposed a ResNet-based spoofing detection framework. Hemlata Tak et al. proposed RawNet2 [9], which models raw waveforms directly and achieves high detection accuracy with low complexity through stacked convolutional blocks and GRU modules.

In terms of feature enhancement, Zhang et al. [10] employed a frequency attention mechanism to enhance the responsiveness of forged regions, significantly improving model stability across different attack types. Wang et al. [6] proposed a multi-channel masking mechanism to highlight deepfake patterns in key frequency bands. Furthermore, to improve model generalization, researchers have introduced more complex architectures into audio modeling. Tang et al. [11] constructed a detection framework based on Graph Neural Networks to capture structural relationships between speech segments. Yu et al. [12] further proposed GraphSpoofNet, combining time-frequency graph convolution with adjacency graph construction, effectively enhancing robustness across attack types and unknown spoofed samples. Simultaneously, the introduction of the Transformer architecture has driven breakthroughs in modeling capability. Gong et al. [13] proposed the Rawformer model, which feeds raw audio sequences into multi-layer Transformers for global modeling, significantly improving detection accuracy. Wang et al. [14] constructed a hybrid multi-scale convolutional and Transformer network to adapt to the extraction of spoofing traces at different temporal scales.

For cross-dataset or cross-speaker scenarios, generalization capability has become a key challenge. Tak et al. [15] proposed a transfer learning strategy, optimizing feature distribution consistency based on domain adaptation ideas, effectively mitigating model overfitting to specific training sets. Chen et al. [16] jointly modeled audio and video forgeries, introducing a multimodal fusion structure that demonstrated superior performance in detecting multi-source synthetic attacks. Additionally, in recent years, some studies have attempted to incorporate fuzzy modeling capabilities. Chen et al. [17] introduced uncertainty estimation to determine forgery confidence, while Zhao et al. [18] employed fuzzy rules to construct classifiers, enhancing detection robustness in noisy environments.

### 2.2. Fuzzy Set Theory Modeling

Fuzzy set theory was first proposed by Zadeh in 1965 [19] to address uncertainties and ambiguities that traditional set theory could not express. Building on this, Atanassov further introduced Intuitionistic Fuzzy Sets (IFS) in 1986 [20], which characterize fuzziness through dual indicators of membership and non-membership degrees. Torra et al. [21] proposed Hesitant Fuzzy Sets (HFS), allowing multiple membership values to express hesitant judgments. Yager et al. [22] further developed Pythagorean Fuzzy Sets (PFS), enhancing expressive capability through square sum constraints. Peng et al. [23] subsequently constructed Pythagorean Hesitant Fuzzy Sets (PHFS), enabling simultaneous modeling of complex hesitation, conflict, and ambiguity, making it one of the most expressive extensions of fuzzy set theory to date.

These fuzzy theories have been widely applied in tasks such as image recognition, medical diagnosis, and multi-attribute decision-making. In the field of image forgery detection, Wang et al. [24] utilized PFS to model the ambiguity in image edge regions, enhancing the response intensity of forged areas. Zhao et al. introduced IFS in medical image segmentation to accurately locate ambiguous pixel boundaries, significantly improving segmentation accuracy. Gupta et al. [25] proposed a PFS-based fuzzy KNN algorithm to enhance the robustness of speech recognition systems in noisy environments. Li et al. [26] combined fuzzy features with attention mechanisms to focus on high-uncertainty regions in images. Zhang et al. [27] achieved promising results in image synthesis and tampering detection tasks by introducing HFS to fuse multi-source fuzzy information.

Fuzzy theory has also seen preliminary applications in the audio domain. Miao et al. [28] proposed fuzzy membership functions to model the uncertainty of speech frequency bands for fuzzy intensity modeling in emotion recognition. Damskägg et al. [29] applied fuzzy classification of spectral bins to audio time stretching, demonstrating the value of fuzzy logic in processing spectral characteristics of audio signals. Chen et al. employed PHFS to define fuzzy semantic labels, enhancing classification stability in low-resource speech recognition tasks. Although these studies demonstrate the feasibility of fuzzy modeling in processing speech data, systematic integration for audio deepfake detection remains relatively scarce.

Currently, a limited number of studies have attempted to incorporate fuzzy set theory into audio detection network structures. Zhou et al. proposed a fuzzy graph convolutional neural network to model fuzzy correlations between speech segments, adapting to audio sequences with varying forgery intensities. These attempts indicate that embedding fuzzy theory into deep detection structures holds significant potential, particularly in modeling asymmetry, uncertainty, and multi-source conflicts.

## 3. Methodology

To address the challenges of ambiguous spoofing traces and environmental noise interference in audio deepfake detection tasks, this chapter proposes a Dual-Path Time-Frequency Attention Network (DPTFAN) based on Pythagorean Hesitant Fuzzy Sets (PHFS) [30]. The core idea is to dynamically model the credibility of audio features using the membership degree (μ), non-membership degree (ν), and hesitancy degree (π) of PHFS, and to construct a dual-path time-frequency attention network endowed with uncertainty awareness. The overall workflow is illustrated in Figure 1.

### 3.1. Data Preprocessing

During the data preprocessing stage, we prepared audio samples from the ASVSpoof 2019 [31] and FoR [32] datasets. In this paper, the audio samples are first divided into multiple short frames, with each frame representing the short-term characteristics of the audio signal. To prevent information loss between frames, the audio signal is segmented into short-time frames with a frame length of 25 ms and a frame shift of 10 ms (50% overlap). The 50% frame overlap ensures signal continuity and facilitates smooth transitions during computation, thereby effectively capturing the time-varying characteristics of the audio signal. Each frame is initially weighted using a Hamming window function to mitigate spectral leakage. The expression of the Hamming window function is as follows:(1)$$W(n)=0.54-0.46\cos(\frac{2\pi n}{N-1})$$
where N is the length of the window function, and n represents the sampling point within the current frame.

The purpose of the Hamming window is to apply weighting to both ends of the signal, causing the amplitude at the edges to gradually attenuate. This avoids spectral leakage caused by frame truncation and ensures the accuracy of spectral calculations. Each frame of the signal, after being weighted by the Hamming window, subsequently undergoes Fast Fourier Transformation (FFT) to convert the time-domain signal into a frequency-domain representation. The formula for FFT is as follows:(2)$$X(f)=\sum_{n=0}^{N-1}x(n)e^{-j2\pi fn/N}$$
where X(f) represents the frequency-domain signal, x(n) denotes the time-domain signal, *N* is the frame length, *f* is the frequency component, and *j* is the imaginary unit.

Through FFT, the frequency components and their amplitude distribution of each frame signal can be calculated, thereby obtaining the spectral information of the frame. However, since the human ear exhibits varying sensitivity to different frequencies, directly using the spectrogram may not adequately capture the characteristics of the audio signal. Therefore, to better align with the perception of the human auditory system, this paper adopts the Mel frequency scale to non-linearly compress the spectrum. The conversion formula for the Mel frequency is as follows:(3)$$f_{mel}=2595\log_{10}(1+\frac{f}{700})$$
where mel is the Mel frequency, and *f* is the linear frequency (Hz).

The Mel scale simulates the human ear’s varying sensitivity to low and high frequencies by non-linearly compressing the frequency axis. Mel frequency transformation helps reduce redundant information in the high-frequency range while preserving critical information in the low-frequency range, making it more aligned with human auditory characteristics. After the Mel frequency transformation, the spectrogram is filtered through a Mel filter bank. Each Mel filter outputs the energy value of a specific frequency range, resulting in the final Mel spectrogram. In this study, we generated Mel spectrograms with a size of 224 × 224 × 3 using a Hanning window with a size of 2048 and a hop length of 512. The number of Mel filters used in the filter bank was 224. The generation of Mel spectrograms is a critical step in the proposed method, as it prepares the audio data for subsequent feature extraction and analysis stages.

### 3.2. Feature-Enhanced Lightweight Fuzzy Branch Network (LFBN)

Traditional deepfake detection models usually assume that features are deterministic; however, in practical scenarios, interferences such as noise and compression introduce uncertainty. To address this issue, this paper proposes a feature enhancement method based on the Pythagorean Hesitant Fuzzy Set (PHFS). By dynamically modeling the credibility and uncertainty of features, this method improves the robustness and interpretability of the model.

Based on the feature representation of preprocessing-generated Mel spectrograms (with a size of 224 × 224 × 3), this method focuses on the key problem of dynamically modeling feature uncertainty and credibility in the deepfake audio identification task. It achieves fuzzy enhancement in the feature space by designing a Lightweight Fuzzy Branch Network (LFBN). Figure 2 shows the overall architecture of this method, and its core processing flow includes the following four key stages.

First, the high-level features of ResNet generate the membership degree (μ) and non-membership degree (ν) through two fully connected branches, which are then mapped to the interval (0, 1) via the Sigmoid function. Subsequently, the PHFS projection layer normalizes μ and ν according to the constraint μ^2^ + ν^2^ ≤ 1, so as to satisfy the mathematical definition of the Pythagorean fuzzy set. Furthermore, the hesitation degree (π) is calculated to quantify the uncertainty and hesitation of features. Finally, μ, ν, and π are concatenated with the original features to form enhanced features, providing the downstream network with a hybrid feature space that integrates both semantic information and uncertainty representation.

Compared with traditional Intuitionistic Fuzzy Sets (IFS), Pythagorean Fuzzy Sets (PFS), or Hesitant Fuzzy Sets (HFS), PHFS has more flexible expressive ability in handling conflicts, hesitation, and uncertainty. It is particularly suitable for addressing degraded or ambiguous forgery traces, thereby significantly improving the robustness and interpretability of the model.

The Mel spectrogram is first processed by a pre-trained ResNet-34 network [33] to extract high-level semantic features, yielding an initial feature representation $X\in R^{B*T*d}$ where B is the batch size, T is the number of time steps (corresponding to the time axis of the Mel spectrogram), and d is the feature dimension. This step aims to capture key patterns in the time-frequency domain of the audio signal. Subsequently, an unnormalized membership degree $\mu^{\prime }$ and non-membership degree $\nu^{\prime }$ are generated through a feature mapping layer composed of two fully connected branches. The calculation methods for the membership and non-membership degrees are as follows:(4)$$\mu^{\prime }=\sigma (W_{\mu }X+b_{\mu })$$(5)$$\upsilon^{\prime }=\sigma (W_{\upsilon }X+b_{\upsilon })$$

Among $W_{\mu },W_{\upsilon }\in R^{d\times l}$ them is the learnable, σ parameter and is the sigmoid function. The dual-branch structure achieves independent modeling of feature credibility and uncertainty through parameter decoupling.

To satisfy the mathematical constraints of Pythagorean fuzzy sets, the generated unnormalized membership $\mu^{\prime }$ and non-membership degrees $\nu^{\prime }$ are subsequently projected into a normalized space via a PHFS constraint layer. The normalization formulas are as follows:(6)$$s=\sqrt{{\mu^{\prime }}^{2}+{\upsilon^{\prime }}^{2}+\varepsilon }$$(7)$$\mu =\frac{\mu^{\prime }}{s}(s>1)$$(8)$$\upsilon =\frac{\upsilon^{\prime }}{s}(s>1)$$

Then, according to the normalized membership degree and non-membership degree, the degree is calculated to quantify the uncertainty of the feature:(9)$$\pi =\sqrt{1-\mu^{2}-\upsilon^{2}}$$

Finally, the membership degree, non-membership degree, and hesitancy degree are concatenated with the original features as additional channels to form the enhanced feature $X^{\prime }\in R^{B\times T\times (d+3)}$. This provides the downstream network with a hybrid feature space that integrates both semantic information and uncertainty representation. By explicitly encoding feature credibility and uncertainty, the model dynamically weighs the contribution of features from different regions when judging spoofed segments.

### 3.3. Dual Pathway Temporal and Spatial Attention Network (DPTFAN)

The Mel spectrogram of audio is essentially a two-dimensional time-frequency matrix with explicit structural properties. The temporal dimension captures the dynamic variations of speech, such as phoneme boundaries, rhythmic patterns, and energy transitions; in contrast, the frequency dimension carries static spectral features, including harmonic structures, formant distributions, and high-frequency artifacts. Extensive research has demonstrated that the forgery cues embedded in these two dimensions exhibit significant complementarity, while there exist intrinsic differences in their statistical characteristics and spatial correlations. If only 2D convolution is employed for holistic modeling, temporal and spectral cues tend to be mixed during the convolution process, which may undermine the model’s capability to independently represent the two types of fine-grained forgery patterns.

Based on this insight, the Mel spectrogram is explicitly decoupled into a Temporal Path (T-Path) and a Frequency Path (F-Path) in the proposed model design. The T-Path adopts a 1D convolution structure along the temporal axis, focusing on capturing the dynamic variations and continuity patterns of speech; the F-Path, on the other hand, retains the complete 2D structure and enhances the modeling of spectral details via 2D convolution and channel attention. These two paths are subsequently fused through the Lightweight Fuzzy Branch Network (LFBN), thereby achieving complementary enhancement of temporal and spectral features. This decoupling-fusion design not only conforms to the time-frequency structure of Mel features but also strengthens the expressiveness of forgery traces across different dimensions, thus forming the design rationale for the dual-path structure proposed in this study.

The proposed Dual-Path Time-Frequency Attention Network (DPTFAN) achieves efficient forgery detection with multi-stage fuzzy information fusion by integrating the time-frequency properties of Mel spectrograms and the feature enhancement capability of the Lightweight Fuzzy Branch Network (LFBN). Leveraging the time-frequency decoupling property of speech signals, the network processes features from the temporal and frequency dimensions separately: the Temporal Path (T-Path) focuses on temporal dynamic patterns (e.g., rhythmic anomalies, phoneme discontinuities), while the Frequency Path (F-Path) captures frequency-domain structural features (e.g., harmonic absence, formant distortion). The dual paths improve detection accuracy through a complementary learning mechanism, and the specific workflow is as follows:

Time Path: The preprocessed Mel spectrogram $I\in R^{224\times (224\times 3)}$ is reshaped into a pseudo-one-dimensional sequence $I_{reshape}\in R^{224\times (224\times 3)}$ (224 time steps, each containing 224 × 3 frequency-domain features), which incorporates concatenated features from all frequency channels. This step isolates the time axis for independent modeling, avoiding interference from frequency-domain information and enabling the model to focus on temporal dynamics. Subsequently, a convolution operation with a kernel size of 5, stride of 2, and padding of 2 is applied along the time axis of the pseudo-one-dimensional sequence to extract temporal features, yielding initial features along the time path. This captures short-term temporal patterns (such as phoneme boundaries and energy variations).

Next, four lightweight residual modules are stacked, each comprising a depthwise separable convolution (with 8 groups), batch normalization, and ELU activation. Each module employs a stride of 2, progressively downsampling the time steps and expanding the feature dimension from 672 to 1344. Residual connections are used to add the input of each module to its output, preventing degradation in deep networks. The purpose of the lightweight residual modules is to balance computational efficiency with audio feature representation, producing the output features along the time path.

The Lightweight Fuzzy Branch Network (LFBN) analyzes the Mel spectrogram and outputs a 14-dimensional vector, representing the importance weight (0~1) of each time step. These weights are applied to the output features of the time path (14 time steps × 1344 dimensions) in a step-wise manner. The membership matrix $\mu^{\prime }$ generated by the LFBN adjusts the weights of key time steps (such as abnormal silent segments) to approach 1, while weights of less important segments approach 0. The calculation method is as follows:(10)$$A_{t}=Soft\max(\frac{QK^{T}}{\sqrt{d_{k}}}⊗\mu^{\prime })\;Q,K\in R^{56\times 256}$$

Among them, d_k_ denotes the dimension of the key (and query) feature vectors in the attention mechanism. To avoid excessively large dot-product similarity (caused by high vector dimensions) which would lead to overly small Softmax gradients, a scaling factor is introduced in the formula to improve the numerical stability of attention calculation. This is consistent with the standard scaled dot-product attention mechanism in the Transformer.

The architecture further incorporates four lightweight residual modules, each featuring depth separable convolution (with 8 groups), batch normalization, and ELU activation. With a stride of 2, the temporal steps are progressively downsampled, expanding the feature dimension from 672 to 1344. Residual connections are applied by summing the inputs and outputs of each module, effectively preventing network degradation in deep layers. These lightweight modules balance computational efficiency with audio feature preservation, ultimately yielding output features along the temporal path.

The Lightweight Fuzzy Branch Network (LFBN) analyzes the Mel frequency spectrum and outputs a 14-dimensional vector representing importance weights (0–1) for each time step. It then applies time-step weighting to the output features $\mu^{\prime }$ (14 time steps × 1344 dimensions). By combining the membership degree matrix generated by LFBN, key time steps (such as abnormal silent segments) are adjusted to have weights close to 1, while secondary segments are assigned weights approaching 0. The calculation method is as follows:

Frequency Path: The preprocessed Mel spectrogram $I\in R^{224\times (224\times 3)}$ is fed into the frequency path. First, a 3 × 3 convolutional kernel is applied to slide along the frequency axis, extracting correlation features between adjacent frequency points (such as harmonic continuity), resulting in feature ${F_{f1}=Conv2D(I}_{reshape})\in R^{112\times 112\times 64}$. Subsequently, a Squeeze-and-Excitation (SE) module is employed, where the squeeze operation performs global average pooling on each channel to produce a 64-dimensional channel descriptor vector. The excitation operation then uses two fully connected layers (with an intermediate dimension of 16) to generate channel weights, enhancing the weights of key frequency bands (4–6 kHz) while suppressing those of noisy bands (such as high-frequency artifacts).

Multi-scale frequency-domain context fusion is achieved by concatenating features processed through spatial pyramid pooling with 4 × 4 max pooling and 2 × 2 average pooling, respectively. This integrates multi-scale frequency-domain contextual information, improving the model’s robustness to variations in frequency band structures. The resulting feature is denoted as ${F_{f2}=Concat(F}_{MAX},F_{avg})\in R^{28\times 28\times 128}$.

The non-membership degree $\nu^{\prime }$, generated by the Lightweight Fuzzy Branch Network (LFBN), identifies noisy or ambiguous regions (values close to 1 indicate suppression is required). The calculation method is as follows:(11)$$F_{f}=F_{f2}⊗(1-\upsilon^{\prime })\in R^{28*28*128}$$

Dual-Path Feature Fusion: First, the final features from the time path are upsampled to the dimension $F_{t}\in R^{28\times 28\times 128}$ using bilinear interpolation to align with the features $F_{f}\in R^{28\times 28\times 128}$ output by the frequency path. Time-frequency cross-attention is employed to fuse the features from the time and frequency paths. The design of the time-frequency cross-attention is as follows:

The query $Q_{t}$ is generated from the time path features, reflecting temporal dynamic patterns,(12)$$Q_{t}=W_{qt}F_{t}\in R^{28*256}$$

The value $V_{f}$ and key $K_{f}$ are generated from the frequency path features, encoding spectral structural information.(13)$$V_{f}=W_{vf}F_{f}\in R^{28*256}$$(14)$$K_{f}=W_{kf}F_{f}\in R^{28*256}$$

The deep hesitancy degree matrix $\pi_{2}\in R^{28\times 28\times 1}$ generated by the Lightweight Fuzzy Branch Network (LFBN) is element-wise multiplied with the attention weights to suppress attention responses in uncertain regions (e.g., background noise), thereby fuzzifying uncertain information:(15)$$Attention=Soft\max(\frac{(Q_{t}K_{f}^{T})⊗\pi_{2}}{\sqrt{256}})V_{f}\in R^{28*28*256}$$

The frequency path features are then weighted and aggregated using the adjusted attention weights, and concatenated with the time path features. The fused features are mapped to a high-dimensional space through a fully connected layer. Finally, a Softmax classifier is applied to output the probabilities of spoofed and genuine classes, completing the end-to-end detection process.

## 4. Experimental Configuration

### 4.1. Data Sets and Indicators

The ASVspoof 2019 dataset is designed to address three major types of speech spoofing attacks—replay, speech synthesis, and voice conversion—and is evaluated through a unified competition framework. The dataset comprises two core scenarios: Logical Access (LA) and Physical Access (PA), each providing distinct datasets tailored to their specific access control challenges. In this study, we primarily focus on the LA scenario, which includes 17 spoofing attacks (A01 to A19) covering a variety of speech synthesis and voice conversion techniques. For example, A01 utilizes WaveNet for high-quality speech synthesis, A02 employs the WORLD vocoder under limited data conditions, and A03 leverages the Merlin toolkit to simplify TTS system construction. A04 highlights natural speech characteristics through waveform concatenation implemented with MaryTTS, while A05 and A06 introduce neural network-based voice conversion methods, employing variational autoencoders and transfer function models, respectively. A07 to A19 further incorporate technologies such as Generative Adversarial Networks (GANs) and neural source-filter models, comprehensively challenging the performance of automatic speaker verification systems.

Additionally, the FoR (Fake or Real) dataset contains over 198,000 speech samples, encompassing speech generated by advanced TTS systems and genuine human speech. The dataset releases four versions: for-original, for-norm, for-2s, and for-rerecorded, aimed at simulating different real-world attack scenarios. The for-original version provides raw audio files from TTS systems and genuine human speech, serving as a benchmark for comparison. The for-norm version ensures volume consistency through audio normalization. The for-2s version clips audio segments to 2 s, making it suitable for short-duration speech analysis. The for-rerecorded version simulates real-world background noise by playing and re-recording the original audio to replicate practical conditions. In our study, we utilized individual versions of these datasets and merged them into a more comprehensive unified dataset for analysis.

On the ASVspoof 2019 LA dataset, to ensure fair comparison with existing studies, we strictly followed the official evaluation protocol and did not introduce additional data augmentation. On the FoR dataset, we adopted two types of lightweight waveform-level augmentations to improve the model’s robustness against environmental variations: Gaussian noise was added, with the Signal-to-Noise Ratio (SNR) randomly sampled within the range of 15–30 dB; Dynamic Range Compression (DRC) was used, implemented based on the Audiomentations library. Each augmentation was randomly applied with a probability of 0.3 during the training phase, and no augmentations were used in the testing phase.

During the evaluation process, we adhered to the standards of the ASVspoof 2019 challenge, primarily using the Equal Error Rate (EER) as the performance evaluation metric. Additionally, we reported accuracy, F1-score, and AUC scores to enable a comprehensive comparative analysis of the proposed model.

### 4.2. Model Parameter Setting

During the preprocessing stage, audio samples are first pre-emphasized with a coefficient of 0.97, then truncated or padded to a fixed length of approximately 4 s (64,600 sample points). We did not apply Voice Activity Detection (VAD) to the audio samples but performed volume normalization with a target amplitude of −20 dB.

The model was trained using the Adam optimizer with β parameters set to [0.9, 0.999]. We employed a stepwise learning rate decay scheduler to accelerate convergence. The initial learning rate was set to 3 × 10^−6^, decaying every 6000 steps with a decay factor (gamma) of 0.1. The batch size was set to 16. All experiments were conducted on four NVIDIA GeForce RTX 4090 GPUs (NVIDIA, Santa Clara, CA, USA). For each configuration, the model was trained for approximately 16,000 steps.

## 5. Results and Analysis

### 5.1. Results on the ASVspoof 2019 Data Set

This experiment was conducted under the Logical Access (LA) scenario of the ASVspoof 2019 dataset to evaluate the performance of the proposed model in detecting various types of spoofed speech attacks. The tests focused specifically on attacks A07 to A19, with detailed evaluations performed for each category, while overall performance was also assessed on the complete dataset. Detection effectiveness was measured using both Accuracy and Equal Error Rate (EER) metrics to ensure a comprehensive validation of the model’s capabilities from multiple perspectives. The experimental results are shown in Figure 3.

In terms of the accuracy metric, the model achieved excellent performance on most attack types. The overall accuracy generally remained at a high level, with the lowest accuracy being 95.62% for the A17 attack, and the highest accuracy reaching 98.94% in the full LA scenario test. As shown in Figure 3a, the accuracy fluctuates slightly across different attack types, demonstrating the model’s good stability and strong generalization ability. The overall results indicate that the model can achieve efficient identification and classification under diverse forged speech conditions, effectively resisting fake audio samples generated by different methods. This paper further conducted a fine-grained evaluation of the model’s performance using the EER metric. As shown in Figure 3b, the EER corresponding to most attack types (A07 to A17) is below 0.41, showing high detection reliability and a low misjudgment rate. However, when facing A18 and A19 attacks, the EER shows a significant upward trend, reaching 1.36 and 0.92 respectively. The relatively high EER of A18 and A19 is mainly due to their adoption of more advanced forgery generation mechanisms. Both types of attacks integrate high-fidelity neural vocoders with multi-stage speech enhancement pipelines, enabling the generated forged speech to exhibit more natural prosodic rhythms in the time domain and smoother spectral structures in the frequency domain. This significantly weakens common forgery artifacts (such as spectral discontinuities, abnormal excitation signals, and sudden energy changes). The aforementioned characteristics simultaneously weaken the forgery cues available to both the temporal path and the frequency path, resulting in relatively weak performance of detection models—including the DPTFAN proposed in this paper—on these two types of attacks. This also explains the high-difficulty feature of A18 and A19 in the LA dataset.

### 5.2. Results on the FoR Dataset

To comprehensively validate the effectiveness and robustness of the proposed Dual-Path Time-Frequency Attention Network (DPTFAN) in audio deepfake detection tasks, we conducted systematic experimental evaluations on all four subset versions of the FoR (Fake or Real) audio dataset—for-original, for-norm, for-2s, and for-rerecorded—as well as on the entire dataset. As a representative and challenging resource in the field of audio deepfake detection, the FoR dataset encompasses a variety of spoofing methods and scenarios, thereby imposing high demands on the model’s capability in time-frequency feature modeling and generalization.

The experimental results, as shown in Figure 4a,b, demonstrate that the DPTFAN model achieves outstanding overall detection performance on the FoR dataset, with an Equal Error Rate (EER) of only 0.02 and an accuracy of 99.4%. This indicates that the model maintains exceptionally high detection accuracy even in complex environments, exhibiting strong classification and discrimination capabilities. Compared to existing methods, the proposed model significantly reduces both false acceptance and false rejection rates in audio deepfake detection tasks, further validating the effectiveness of its joint time-frequency modeling mechanism.

Upon detailed analysis of the four subsets of the FoR dataset, we observed that the DPTFAN model achieves excellent performance across all subset versions, demonstrating high stability and robustness. On the “for-original” subset, the model attained an EER of 0.06 and an accuracy of 98.2%, indicating its strong discriminative capability even on unmodified raw audio data. In the “for-norm” subset (comprising normalized spoofed samples), the model further exhibited superior performance, with an EER of only 0.05 and an accuracy of 99.1%, demonstrating that DPTFAN can maintain efficient detection even after feature compression and amplitude normalization. For the more challenging “for-2s” subset (where each audio segment is truncated to two seconds), DPTFAN still maintained strong performance, achieving an EER of 0.10 and an accuracy of 98.6%. This verifies its adaptability in short-duration spoofed audio detection tasks, indicating that the model can not only extract discriminative features from long-term dependencies but also accurately identify spoofed samples under time-constrained conditions, showcasing robust temporal compression resilience.

On the “for-rerecorded” subset, which involves re-recording spoofed samples to simulate audio degradation in real-world transmission scenarios, the detection difficulty increases significantly. Despite this, DPTFAN maintained an EER of 1.85 and an accuracy of 96.9%. When confronted with multiple influencing factors such as channel interference, background noise, and device resampling, the model still demonstrated strong robustness and considerable anti-interference capability. Compared to traditional models, which often experience significant performance degradation on this subset, DPTFAN maintains an acceptable error margin in such degraded audio scenarios, highlighting its potential applicability in real-world complex environments for deepfake audio detection tasks.

### 5.3. Model Comparison Results

To comprehensively evaluate the effectiveness of the proposed Pythagorean Hesitant Fuzzy Set (PHFS)-based Dual-Path Time-Frequency Attention Network (DPTFAN), this study conducted systematic comparative experiments on two representative audio deepfake detection datasets: ASVspoof 2019 LA and the comprehensive FoR dataset. Multiple representative audio deepfake detection models were selected as baselines, covering a range of architectures from traditional convolutional networks (e.g., VGG16, CNN-LSTM) to state-of-the-art lightweight and Transformer-based models (e.g., MobileNet-V3, ConvNeXt-Tiny, MFCMNet). The experimental results are presented in Table 1.

As clearly shown in Table 1, the performance of audio deepfake detection has significantly improved with the evolution of model architectures. The traditional CQCC-GMM model achieved only 79.85% accuracy on the ASVspoof 2019 dataset, while advanced models based on Transformers and attention mechanisms, such as Rawformer and GraphSpoofNet, improved the accuracy to 97.03% and 97.20%, respectively. This demonstrates the superior capability of deep learning architectures in capturing complex spoofed speech features.

As simple fuzzy enhancement models, CNN + IFS and CNN + PFS improved the accuracy from the baseline CNN’s 86.20% to 87.60% and 88.80% respectively, while reducing the EER from 10.54 to 9.80 and 9.10 respectively. Similar performance improvements were observed on the FoR dataset (accuracy increased from 82.10% to 83.50%, and EER decreased from 12.50 to 11.80). These results indicate that simple fuzzy enhancement can improve model performance, but the magnitude of improvement is limited.

In contrast, the proposed DPTFAN model introduces Pythagorean Hesitant Fuzzy Sets (PHFS) to enable more refined modeling of uncertainty in time-frequency features. By combining the Dual-Path Time-Frequency Attention Network (DPTFAN) to simultaneously extract fine-grained features from both the time and frequency domains, the model effectively enhances its discriminative sensitivity to spoofed speech. On the ASVspoof 2019 LA dataset, DPTFAN achieved an accuracy of 98.94%, which is 1.74 percentage points higher than the suboptimal model, GraphSpoofNet. Moreover, the Equal Error Rate (EER) was significantly reduced to 0.35, far surpassing Rawformer’s 3.95. On the comprehensive FoR dataset, the model performed even more prominently, achieving an accuracy of 99.40% and an EER of only 0.02, demonstrating a significant improvement over the second-best model, Rawformer. The F1-score and AUC metrics also reached 0.978 and 0.991 (on ASVspoof 2019) and 0.995 and 0.998 (on FoR), respectively, reflecting the model’s comprehensive advantages in detection accuracy and robust discriminative capability.

This performance superiority can be attributed to the expressive power of PHFS in handling data ambiguity and uncertainty, as well as the synergistic effect of the DPTFAN structure along the dual time and frequency paths. These components effectively capture subtle anomalous information in spoofed speech, significantly enhancing the model’s discriminative power and generalization ability.

To further validate the contribution of each core component (the PHFS fuzzy enhancement mechanism and the DPTFAN dual-path attention structure) to the overall detection performance, ablation studies were conducted by combining a baseline CNN network with the LFBN module, the Time Path (T-Path), and the Frequency Path (F-Path) in different configurations. The performance of these variants was compared on both the ASVspoof 2019 LA and FoR datasets. The experimental results are presented in Table 2.

As evidenced in Table 2, the baseline CNN model achieves an accuracy of 79.85% and an Equal Error Rate (EER) of 15.32 on the ASVspoof 2019 dataset, while on the FoR dataset, it attains 80.45% accuracy and a 13.82 EER, indicating relatively weak detection performance. After incorporating the LFBN module, the model’s performance improves significantly, with accuracy on ASVspoof 2019 rising to 86.20% and the EER decreasing to 10.54. This demonstrates that the PHFS-based fuzzy enhancement effectively improves the model’s perception of subtle details in spoofed audio.

The Frequency Path (CNN + F-Path) and Time Path (CNN + T-Path) each exhibit advantages in modeling static spectral features and temporal dynamics, respectively, increasing accuracy to 89.55% and 90.04%, while reducing EER to 8.67 and 8.31. When both F-Path and T-Path are jointly integrated (CNN + F-Path + T-Path), the model achieves further improved performance on both datasets, with accuracy reaching 92.35% and 86.10%, and EER further decreasing to 6.88 and 9.35, validating the complementary nature of joint time-frequency modeling.

Additionally, combining LFBN with either F-Path or T-Path (e.g., CNN + LFBN + F-Path or CNN + LFBN + T-Path) also yields consistent performance gains. Ultimately, the complete DPTFAN model, which integrates PHFS with the dual-path attention mechanism, achieves an accuracy of 98.94% and an EER of 0.35 on ASVspoof 2019. Its performance on the FoR dataset is even more exceptional, with an accuracy of 99.40% and an EER of only 0.02, significantly surpassing all comparative configurations. This fully demonstrates the synergistic effects and effectiveness of the core components in spoofed audio detection.

To further analyze the role of each component in the Pythagorean Hesitant Fuzzy Set (PHFS), we conducted ablation experiments on μ (membership degree), ν (non-membership degree), and π (hesitation degree) individually and in combinations, as shown in Table 3.

The results show that when only the μ component is used, the accuracy on the ASVspoof 2019 LA dataset is 92.50% with an EER of 6.80, and on the FoR dataset, the accuracy is 88.00% with an EER of 8.50. When only the ν component is used, the accuracy slightly increases to 92.80% and the EER decreases to 6.50 on the ASVspoof 2019 LA dataset; on the FoR dataset, the accuracy reaches 88.20% with an EER of 8.30. These findings indicate that each component can provide a certain improvement to the performance of the baseline CNN. When μ and ν are used in combination, the accuracy on the ASVspoof 2019 LA dataset rises to 95.20% with an EER of 3.80, and on the FoR dataset, the accuracy is 91.50% with an EER of 5.10. This demonstrates that combined modeling can further enhance the feature discrimination capability. The complete PHFS (μ + ν + π) achieves the optimal performance on both datasets: on the ASVspoof 2019 LA dataset, the accuracy is 98.94% with an EER of only 0.35; on the FoR dataset, the accuracy reaches 99.40% with an EER of 0.02. Meanwhile, the F1 score and AUC also reach their highest values simultaneously. This indicates that simultaneously modeling the membership degree, non-membership degree, and hesitation degree plays a crucial role in capturing the uncertainty of audio features, and can significantly improve the accuracy and robustness of fake audio detection. To fairly compare the detection capabilities of different models, we counted the number of parameters and FLOPs (Floating-Point Operations Per Second) for all baseline models (such as VGG16, MobileNet-V3, AASIST, Rawformer, GraphSpoofNet, etc.) as well as the proposed DPTFAN method. The results are shown in Table 4.

As can be seen from Table 4, the overall scale of DPTFAN is at a medium level among current mainstream audio deepfake detection models, and it does not rely on a significant increase in the number of parameters or computational complexity to achieve performance improvement. Specifically, DPTFAN has 18.90 million parameters and 12.60 billion FLOPs, which is lower than most Transformer-based or graph-structured models, such as Rawformer (32.80M parameters, 25.40G FLOPs) and GraphSpoofNet (27.60M parameters, 14.20G FLOPs), and also significantly smaller than high-complexity models like ConvNeXt-Tiny (28.60M parameters, 23.10G FLOPs). Meanwhile, compared with structured attention networks such as AASIST (22.40M parameters, 9.70G FLOPs), DPTFAN provides stronger detection capability while maintaining lower computational overhead. In contrast, although lightweight models like MobileNet-V3 (3.10M parameters, 0.42G FLOPs) have a smaller scale, they struggle to capture complex forgery traces in practical detection tasks; traditional CNNs or WaveletCNNs, despite having a moderate scale (5–7 million parameters), are limited by their single-path or local modeling capabilities, resulting in performance far inferior to the dual-path + fuzzy set structure proposed in this study. It is worth noting that although the parameter scale of DPTFAN is higher than that of the most basic CNNs or lightweight architectures, it is still significantly lower than that of the most advanced deep attention models. Nevertheless, DPTFAN achieves the highest accuracy and the lowest EER on both the ASVspoof 2019 LA and FoR datasets. This result fully indicates that the performance advantage of DPTFAN does not come from the stacking of model capacity, but mainly from the structural improvement brought by its dual-path time-frequency modeling mechanism and PHFS-based fuzzy uncertainty enhancement. That is, it achieves detection capability far exceeding that of high-complexity models with a relatively moderate model scale, demonstrating excellent performance-complexity balance and the effectiveness of structural design. To further rule out the possibility that “performance improvement stems from an increase in the number of parameters”, we counted the parameter scale and FLOPs of various sub-models involved in the ablation experiments (such as CNN, CNN + LFBN, CNN + T-Path, CNN + F-Path, etc.). The specific results are shown in Table 5.

As can be seen from Table 5, the introduction of each module in the ablation experiments has distinct impacts on the model’s parameter count and computational complexity, but the overall performance improvement is not simply driven by an increase in the number of parameters. The baseline CNN has 5.30M parameters and 1.02G FLOPs; as a lightweight baseline model, its detection capability is relatively limited. After integrating the LFBN module, the number of parameters slightly increases to 5.68M and the FLOPs rise to 1.14G, while the performance is significantly improved. This indicates that the PHFS-based fuzzy enhancement can effectively capture the uncertainty information in audio features. When the F-Path or T-Path is added individually, the parameter counts are 6.21M and 6.05M respectively, with FLOPs of 1.42G and 1.37G respectively. These lightweight paths significantly enhance the model’s discriminative ability for frequency-domain or temporal features. When both F-Path and T-Path are introduced simultaneously, the number of parameters increases to 7.48M and the FLOPs reach 1.89G, enabling the model to jointly utilize time-frequency information and further improve performance. When the LFBN is combined with a single path (CNN + LFBN + F-Path or CNN + LFBN + T-Path), the number of parameters increases significantly to 13.62M and 14.48M, while the increase in FLOPs is limited (1.56G and 1.51G). This suggests that the additional parameters mainly come from the deep feature extraction and fuzzy enhancement modules. The final complete DPTFAN model has 18.90M parameters and 12.60G FLOPs, which is much higher than the basic ablation models. However, its performance improvement is not driven by simple parameter stacking, but by structural optimizations, including the dual-path time-frequency attention mechanism, PHFS-based fuzzy feature enhancement, and collaborative interaction between modules. Overall, the statistical results in Table 4 fully demonstrate that the performance improvement mainly relies on structural design and uncertainty modeling rather than a simple increase in parameters. At the same time, it also reflects that under the premise of maintaining reasonable computational complexity, DPTFAN can give full play to the advantages of dual-path and fuzzy modeling, achieving better detection performance for audio deepfakes.

### 5.4. Generalization and Complex Forgery Type Testing

To further verify the generalization ability of the DPTFAN model, we used the model weights trained on the ASVspoof 2019 LA and FoR datasets to perform inference on a self-built fake audio dataset (VCTK-based Deepfake Dataset, VCTK-DF). This dataset is based on the VCTK speech dataset, with corresponding fake audios generated through different TTS/VC systems. It also includes partial noise interference and codec variation conditions to simulate the scenario of unknown fake audios in real-world environments. The test results are shown in Table 6.

The experimental results show that the accuracy and EER of the model on the new dataset are slightly lower than those on the original training dataset, but still remain at a high level. For instance, on speech samples generated by unseen TTS systems, the accuracy is 94.80% with an EER of 1.25; on speech samples generated by unseen VC systems, the accuracy reaches 94.50% with an EER of 1.30; under background noise interference, the accuracy is 93.90% with an EER of 1.50; and under codec variation conditions, the accuracy is 94.10% with an EER of 1.40. These results indicate that DPTFAN can still effectively capture the features of fake audio under the scenarios of unseen generation models and complex environments, demonstrating excellent generalization and robustness.

## 6. Summary

This paper proposes a novel detection framework, DPTFAN, based on a dual-path attention structure and fuzzy feature enhancement mechanism for the task of audio deepfake detection. The model consists of three core components: first, an LFBN module based on the Pythagorean Hesitant Fuzzy Set (PHFS) is introduced to perform fuzzy enhancement on input speech features, thereby improving the ability to perceive fuzzy regions and forged traces at boundaries; second, a dual-path attention mechanism is constructed, with a Frequency Path (F-Path) and a Temporal Path (T-Path) designed respectively to achieve dual modeling of the frequency-domain patterns and temporal dynamic changes of forged speech; finally, the discrimination of forged audio is completed through path fusion and a fully connected classifier. Experiments on two authoritative speech forgery detection datasets, ASVspoof 2019 LA and FoR, demonstrate that the proposed DPTFAN model significantly outperforms existing methods in multiple metrics. Meanwhile, DPTFAN effectively improves the detection accuracy of audio deepfakes while maintaining model simplicity. At the same time, we have also identified the model’s limitations: DPTFAN is mainly trained and evaluated on standard datasets and under controlled conditions. In reality, attackers may adopt more advanced or adaptive strategies, such as diffusion model-based TTS or personalized voice cloning. Under such highly adaptive attack conditions, the model may experience performance degradation. Additionally, the model currently focuses on single-language speech forgery, and its generalization ability for cross-lingual or cross-domain tasks (e.g., emotional speech, singing voice synthesis) remains to be further verified. Furthermore, chaotic systems can impose highly controllable nonlinear perturbations on audio signals in the time-frequency domain while maintaining speech intelligibility. For audio deepfake detection models, such perturbations can simulate complex interferences that may occur in real-world environments, such as noise superposition, transmission distortion, or nonlinear features introduced by generation models. Therefore, using chaotic perturbations as a robustness evaluation benchmark helps verify the model’s stability under highly complex or unknown conditions. In future work, we will attempt to conduct systematic testing on DPTFAN using audio perturbations based on memristors or other chaotic systems to further evaluate its generalization and robustness. In response to these limitations, future work will consider optimizing the structural design of the PHFS fuzzy set, introducing a cross-lingual forgery recognition mechanism, and combining adversarial training with domain adaptation techniques to further expand the experimental scope and enhance the model’s robustness in complex environments and under stronger attack conditions.

## Figures and Tables

**Figure 1 sensors-25-07608-f001:**
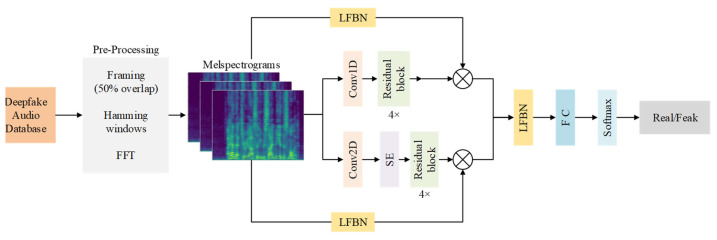
Overall network flow chart.

**Figure 2 sensors-25-07608-f002:**
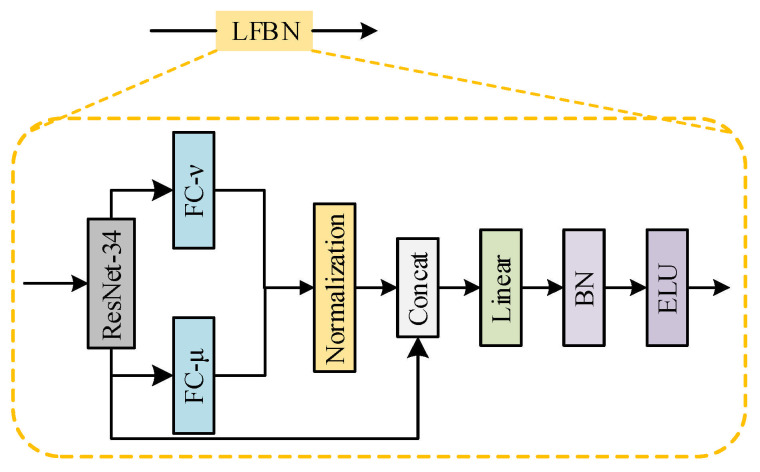
Lightweight Fuzzy Branch Network Structure Diagram.

**Figure 3 sensors-25-07608-f003:**
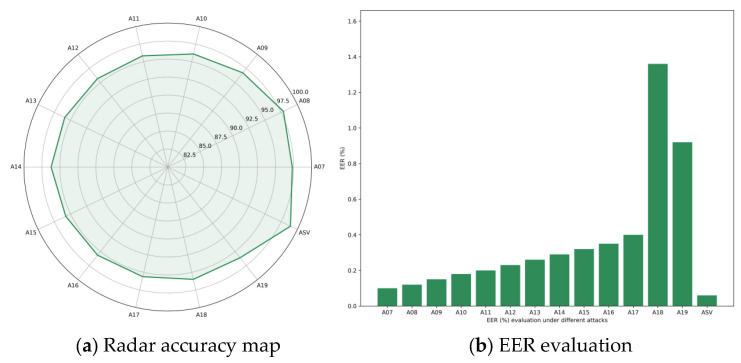
Test results of the ASVspoof 2019 dataset.

**Figure 4 sensors-25-07608-f004:**
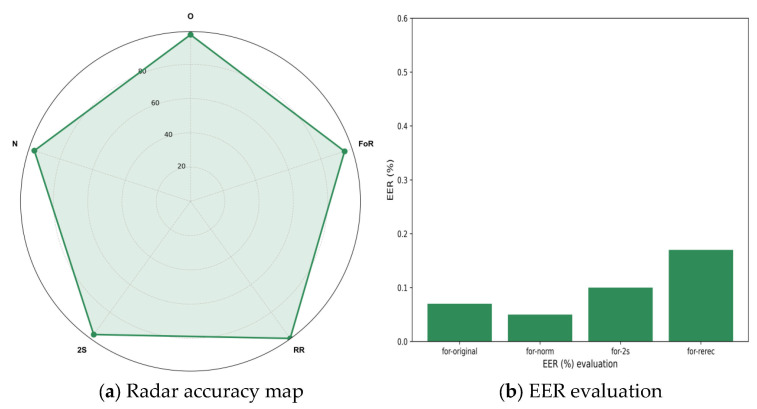
Test results of the FoR dataset.

**Table 1 sensors-25-07608-t001:** Performance Comparison of Different Models on ASVspoof 2019 LA and FoR Datasets.

Model	ASVspoof 2019				FoR			
	Acc (%)	EER	F1	AUC	Acc (%)	EER	F1	AUC
CQCC-GMM [34]	79.85	15.32	0.812	0.860	71.34	18.90	0.744	0.812
CNN	86.20	10.54	0.864	0.917	80.45	13.82	0.816	0.875
VGG16	89.55	8.67	0.887	0.935	82.13	12.01	0.835	0.892
MobileNet-V3 [35]	91.40	7.93	0.901	0.944	84.25	10.87	0.851	0.906
CNN-LSTM [36]	90.68	8.25	0.894	0.938	83.20	11.43	0.842	0.899
WaveletCNN [37]	91.73	7.75	0.905	0.946	85.04	10.25	0.862	0.911
ConvNeXt-Tiny [38]	94.88	5.93	0.935	0.963	88.20	8.21	0.896	0.934
ABC-CapsNet [39]	94.15	6.15	0.928	0.959	87.60	8.90	0.890	0.930
AASIST [40]	96.17	4.73	0.950	0.974	89.45	5.63	0.906	0.940
Rawformer [41]	97.03	3.95	0.961	0.982	92.12	4.41	0.928	0.956
GraphSpoofNet [42]	97.20	3.87	0.963	0.984	91.90	3.53	0.926	0.954
CNN + IFS	87.60	9.80	0874	0.925	82.10	12.50	0.828	0.889
CNN + PFS	88.80	9.10	0.882	0.932	83.50	11.80	0.835	0.897
**DPTFAN**	**98.94**	**0.35**	**0.978**	**0.991**	**99.40**	**0.02**	**0.995**	**0.998**

**Table 2 sensors-25-07608-t002:** Comparative Results of Ablation Experiments.

Model	ASVspoof 2019				FoR			
	Acc (%)	EER	F1	AUC	Acc (%)	EER	F1	AUC
CNN	79.85	15.32	0.812	0.860	80.45	13.82	0.816	0.875
CNN + LFBN	86.20	10.54	0.864	0.917	81.86	12.31	0.841	0.898
CNN + F-Path	89.55	8.67	0.887	0.935	82.13	12.01	0.835	0.892
CNN + T-Path	90.04	8.31	0.863	0.927	84.64	11.62	0.828	0.889
CNN + F-Path +T-Path	92.35	6.88	0.913	0.951	86.10	9.35	0.867	0.914
CNN + LFBN +F-Path	91.40	7.93	0.901	0.944	84.25	10.87	0.851	0.906
CNN + LFBN +T-path	90.60	7.45	0.895	0.940	83.35	10.22	0.845	0.901
**DPTFAN**	**98.94**	**0.35**	**0.978**	**0.991**	**99.40**	**0.02**	**0.995**	**0.998**

**Table 3 sensors-25-07608-t003:** Comparison of the Contribution of PHFS Components.

Model	ASVspoof 2019				FoR			
	Acc (%)	EER	F1	AUC	Acc (%)	EER	F1	AUC
μ only	92.95	6.80	0.910	0.950	88.00	8.50	0.870	0.915
ν only	92.80	6.50	0.912	0.952	88.20	8.30	0.872	0.917
μ + ν	95.20	3.80	0.940	0.970	91.50	5.10	0.910	0.950
μ + ν + π	98.94	0.35	0.978	0.991	99.40	0.02	0.995	0.998

**Table 4 sensors-25-07608-t004:** Comparison of Model Parameters and FLOPs.

Model	Params (M)	FLOPs (G)
CQCC-GMM	0.12	0.05
CNN	5.30	1.02
VGG16	14.70	15.30
MobileNet-V3	3.10	0.42
CNN-LSTM	7.80	2.40
WaveletCNN	6.40	1.35
ConvNeXt-Tiny	28.60	23.10
ABC-CapsNet	11.20	3.60
AASIST	22.40	9.70
Rawformer	32.80	25.40
GraphSpoofNet	27.60	14.20
**DPTFAN (Ours)**	**18.90**	**12.60**

**Table 5 sensors-25-07608-t005:** Comparison of Parameters and FLOPs in Ablation Experiments.

Ablation Model	Params (M)	FLOPs (G)
CNN	5.30	1.02
CNN + LFBN	5.68	1.14
CNN + F-Path	6.21	1.42
CNN + T-Path	6.05	1.37
CNN + F-Path+ T-Path	7.48	1.89
CNN + LFBN + F-Path	13.62	1.56
CNN + LFBN + T-Path	14.48	1.51
**DPTFAN (Full)**	**18.90**	**12.60**

**Table 6 sensors-25-07608-t006:** Comparative Results on the Generalization Ability of the DPTFAN Model.

Test Condition	Dataset	Accuracy (%)	EER
Unseen TTS	VCTK-DF TTS	94.80	1.25
Unseen VC	VCTK-DF VC	94.50	1.30
Background Noise	VCTK-DF noisy	93.90	1.50
Codec Variation	VCTK-DF codec	94.10	1.40
Challenging Attack	A18	95.62	1.36
Challenging Attack	A19	96.50	0.92

## Data Availability

The data presented in this study are available at http://www.asvspoof.org/index2019.html (accessed on 5 November 2025).

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
