# Peer review of "Audio Deepfake Detection via a Fuzzy Dual-Path Time-Frequency Attention Network"

_sensors, 2025, doi:10.3390/s25247608_

Round 1

Reviewer 1 Report

Comments and Suggestions for Authors

This paper introduces a fuzzy dual-path time–frequency attention network for audio deepfake detection. The proposed model consists of two complementary paths that process Mel spectrograms as multi-channel 1D signals and as 2D images, respectively, along with a fuzzy branch designed to handle uncertainty introduced by noise and compression. The model is evaluated on two datasets, with comparisons to existing approaches demonstrating its superiority. Finally, ablation studies validate the contribution of the dual-path design and the fuzzy branch network.

Overall, the paper is well structured and easy to follow. However, I have several concerns, particularly regarding the motivation and the reported results. My detailed comments are as follows:

1. Section 1 (Introduction): The authors claim that “existing methods overlook the feature uncertainty introduced by noise and compression,” which seems misleading. Machine learning models—especially neural networks—are inherently designed to generalize, and uncertainty or noise is always present during both training and inference. Any ML model is non-deterministic by nature, and handling uncertainty is a standard part of model design. While some models may overfit and thus become less robust to noise, preventing overfitting is a common practice across existing approaches. The authors should more clearly articulate what specific challenges existing methods fail to address.

2. There are no preliminary experiments or analyses to justify why combining the 1D and 2D signal models leads to improved performance. The authors should provide an explanation or rationale for how this dual-path structure was chosen. Without such clarification, it appears to be based on trial and error rather than a well-founded design choice.

3. The model name "time-frequency attention network" is somewhat misleading, as the input consists solely of Mel spectrograms, with no time-domain data used. The authors design convolutional kernels for different dimensions of the Mel spectrograms, rather than directly incorporating time-domain signals.

4. In Section 5.3 (Model Comparison Results), it is unclear whether the observed improvement of DPTFAN over existing models is due to the structural optimization provided by the dual-path and fuzzy branch, or simply because of a larger model scale with more parameters and neurons, enhancing its fitting capability. To address this, the authors should report the scale of all models (e.g., number of parameters, number of layers) for comparison.

5. Building on my previous comment, the authors should also evaluate the model scale for the models used in the ablation experiments.

6. I could not find a systematic description of how Pythagorean Hesitant Fuzzy Sets are generated or their intended purpose in the model.

Reviewer 2 Report

Comments and Suggestions for Authors

For details, please refer to the PDF file.

Reviewer 3 Report

Comments and Suggestions for Authors

see the attachment

Round 2

Reviewer 1 Report

Comments and Suggestions for Authors

The manuscript has been adequately revised to address my earlier comments. Nevertheless, the bottom lines in Tables 3, 5, and 6 are missing, which results in visual blending with the table content. Please add the bottom borders.

Author Response

Thank you for your feedback. Please refer to the attachment.

Reviewer 3 Report

Comments and Suggestions for Authors

Accept in present form

Author Response

Thank you for your feedback.